# High-Frequency, Low-Intensity Pulsed Electric Field and N-Acetylcysteine Synergistically Protect SH-SY5Y Cells Against Hydrogen Peroxide-Induced Cell Damage In Vitro

**DOI:** 10.3390/antiox14101267

**Published:** 2025-10-21

**Authors:** Fang-Tzu Hsu, Yu-Yi Kuo, Chih-Yu Chao

**Affiliations:** 1Laboratory for Medical Physics & Biomedical Engineering, Department of Physics, National Taiwan University, Taipei 106319, Taiwan; tsyr8924503@phys.ntu.edu.tw (F.-T.H.); yykuo@phys.ntu.edu.tw (Y.-Y.K.); 2Molecular Imaging Center, College of Medicine, National Taiwan University, Taipei 100233, Taiwan; 3Graduate Institute of Applied Physics, Biophysics Division, National Taiwan University, Taipei 106319, Taiwan

**Keywords:** oxidative stress, neuroprotection, N-acetylcysteine, electric field stimulation, SH-SY5Y cells

## Abstract

Oxidative stress plays an important role in the progression of neurodegenerative diseases (NDDs), and N-acetylcysteine (NAC) has gained attention as a potential agent due to its antioxidant capabilities. This study investigated the synergistic neuroprotective effects of combining NAC with non-contact, high-frequency, low-intensity pulsed electric field (H-LIPEF) stimulation on SH-SY5Y human neuronal cells subjected to hydrogen peroxide (H_2_O_2_)-induced oxidative damage. It was found that after SH-SY5Y cells were pretreated with NAC and exposed to H-LIPEF stimulation, the oxidative stress of cells was reduced in the subsequent treatment with H_2_O_2_. The results showed that the combined NAC and H-LIPEF treatment significantly improved cell viability and more effectively reduced mitochondrial apoptosis. Mechanistic analyses revealed that the combination substantially decreased levels of superoxide and intracellular H_2_O_2_, which was associated with enhanced activation of the phosphorylated Akt (p-Akt)/nuclear factor erythroid 2-related factor 2 (Nrf2)/superoxide dismutase type 2 (SOD2) signaling pathway. Furthermore, the treatment reduced the accumulation of 8-oxo-2′-deoxyguanosine triphosphate (8-oxo-dG) accumulation and elevated MutT homolog 1 (MTH1) expression, indicating a protective effect against oxidative DNA damage. These results suggest that H-LIPEF enhances the neuroprotective efficacy of low-dose NAC, highlighting the potential of this combination approach as a new therapeutic strategy for the treatment of NDDs.

## 1. Introduction

Neurodegenerative diseases (NDDs) encompass a range of disorders characterized by the progressive degeneration of the nervous system. These diseases, which include Alzheimer’s disease (AD) and Parkinson’s disease (PD), often lead to a gradual decline in cognitive or motor functions. The increasing prevalence of NDDs is a significant concern, particularly in light of the aging global population. Current therapeutic options for these diseases are limited; traditional pharmacological interventions, such as cholinesterase inhibitors for AD and Levodopa for PD, primarily focus on managing symptoms rather than halting or reversing disease progression [1]. Consequently, there is an urgent need for a clear understanding of the underlying mechanisms of these diseases and the identification of more effective treatment strategies.

Over the past few decades, researchers have shown that reactive oxygen species (ROS) are pivotal in NDDs [2,3]. Multiple factors contribute to the overproduction of ROS in neuronal cells, including mitochondrial dysfunction [4,5], decreased antioxidant and/or oxidoreductase defenses [1], and the accumulation of misfolded proteins [6,7]. Elevated levels of ROS can lead to DNA damage, lipid peroxidation, and protein oxidation, which subsequently disrupt the regulation of neurotrophic factors and neurotransmitters [1]. Furthermore, increased ROS can impair mitochondrial function and promote the aggregation of misfolded proteins, creating a feedback loop that exacerbates ROS generation and ultimately results in neuronal degeneration [8,9]. Therefore, the development of pharmacological agents or therapeutic approaches aimed at reducing ROS levels presents a promising avenue for the treatment of NDDs. One important pathway that defends against oxidative stress in neurons is the phosphorylated Akt (p-Akt)/nuclear factor erythroid 2-related factor 2 (Nrf2) axis. Phosphorylation of Akt promotes cell survival and has been shown to facilitate stabilization and nuclear translocation of the transcription factor Nrf2 [10]. Once in the nucleus, Nrf2 upregulates the expression of antioxidant enzymes [11,12]. Several studies have shown that interventions that enhance the p-Akt/Nrf2 pathway confer protection against oxidative stress in neurons, which strongly suggests that this pathway is a viable therapeutic target [13,14].

N-acetylcysteine (NAC), a dietary supplement, has garnered attention for its neuroprotective effects due to its antioxidative and anti-inflammatory abilities [15]. As a precursor to L-cysteine, NAC is integral to the intracellular synthesis of glutathione, a vital antioxidant that protects cells against oxidative stress [16]. NAC has also been reported to activate the Akt pathway [17,18], which may contribute to cytoprotection. Several clinical trials have assessed the efficacy of NAC in mitigating the effects of NDDs such as AD [19,20,21] and PD [22,23,24]. The cumulative findings suggest that NAC may serve as a potential adjuvant therapy for NDDs, exhibiting rare side effects and offering the possibility of combinational treatment with other drugs. However, the ability of NAC to effectively penetrate the blood–brain barrier (BBB) and its impact on neuronal cells remain uncertain [25,26,27,28]. Additionally, similar to other antioxidants, NAC exhibits low oral bioavailability (<10%) [29,30], which may limit its therapeutic effectiveness and necessitate administration at relatively high doses [2]. Thus, enhancing the efficacy and potency of NAC is a critical area of investigation.

In addition to pharmacological treatments, electric current stimulation therapies have garnered increasing attention in recent years. Specifically, research indicates that such stimulation can enhance antioxidant defense mechanisms by promoting the expression of Nrf2 and its downstream antioxidant enzymes, including superoxide dismutase type 2 (SOD2) [31,32]. Furthermore, a variety of non-invasive electric current stimulation techniques, such as transcranial direct current stimulation and transcranial alternating current stimulation, have been investigated for their therapeutic potential in NDDs. Evidence from several mouse models and clinical trials suggests that these non-invasive modalities can enhance cognitive and motor functions in individuals with NDDs [33,34,35,36]. Therefore, by augmenting antioxidant defenses, electric current therapies have the potential to serve as a valuable approach to treating NDDs.

Despite the therapeutic potential of using electric current stimulation in treating diseases, several challenges remain. For example, when administered via direct contact, the electric current traverses from the electrode through the tissues, potentially resulting in adverse sensations such as itching, tingling, and skin lesions [37,38]. Hence, in our previous research, we developed a low-intensity pulsed electric field (LIPEF) stimulation device that functions in a non-contact manner [39,40,41]. Our research team has applied LIPEF to in vitro NDD models, demonstrating its capacity to activate crucial molecular pathways and produce significant antioxidative effects. For instance, our team found that LIPEF at a frequency of 2 Hz can protect NSC-34 motor neurons from oxidative stress by modulating the rho-associated protein kinase (ROCK) and downstream Akt protein [39]. Furthermore, we discovered that high-frequency, low-intensity pulsed electric field (H-LIPEF) at 200 Hz can protect SH-SY5Y human neuronal cells from the detrimental effects of hydrogen peroxide (H_2_O_2_) and β-amyloid by activating the extracellular signal-regulated kinase (ERK) pathway [40]. In addition to utilizing LIPEF alone, we showed a capability to combine it with other drugs to exhibit a more pronounced therapeutic effect [39,41]. In sum, these studies suggest that LIPEF has the potential to be a safe treatment modality with significant neuroprotective potential, which may be effectively combined with conventional therapies to achieve a synergistic antioxidative effect.

In this paper, we first study the neuroprotective effects of combining NAC with H-LIPEF stimulation against oxidative stress induced by H_2_O_2_ in SH-SY5Y cells. The results show that the combined treatment of NAC and H-LIPEF is more effective in alleviating mitochondrial apoptosis induced by H_2_O_2_ than either NAC or H-LIPEF treatment alone. Further investigation into the underlying mechanisms suggests that NAC and H-LIPEF act synergistically to reduce the H_2_O_2_-induced increase in superoxide levels, likely through the activation of the p-Akt/Nrf2/SOD2 pathway, which facilitates the conversion of superoxide (O_2_^·–^) into the less harmful H_2_O_2_. Additionally, the experiments show that the combination treatment promotes the further decomposition of H_2_O_2_ into water, resulting in a more substantial reduction in cellular oxidative stress. Furthermore, we demonstrate that the combined NAC and H-LIPEF treatment effectively mitigates oxidative DNA damage by enhancing the expression of the MutT homolog 1 (MTH1) protein. These findings indicate that the combination of NAC and H-LIPEF may represent a safe and promising novel therapeutic approach for NDDs.

## 2. Materials and Methods

### 2.1. Experimental Setup for Cell Exposure to Non-Contact H-LIPEF

The H-LIPEF stimulation apparatus was employed to expose SH-SY5Y cells to non-contact capacitive coupling electrical stimulation, as previously described in our research and illustrated in Figure 1A [40]. The cells seeded in culture wells were placed between two parallel and flat copper electrodes. An electric signal was generated using a function generator (Agilent 33220A, Agilent Technologies, Santa Clara, CA, USA) and subsequently amplified by a voltage amplifier (PZD 700, Trek, Inc., Medina, NY, USA). The electric signal was applied to the electrodes to establish an electric field across the biological sample. In this study, consecutive pulses with a voltage drop of 90 V, a frequency of 200 Hz, and a pulse duration of 2 ms (Figure 1B) were administered between the electrodes, which were separated by a distance of 9 cm. The cells were maintained in a humidified incubator with 5% CO_2_ and 95% air at 37 °C during the continuous exposure to H-LIPEF. The parameters of H-LIPEF utilized in this study adhere to the safety standards established by both the International Commission on Non-Ionizing Radiation Protection and the Institute of Electrical and Electronics Engineers.

### 2.2. Cell Culture

The SH-SY5Y cells, purchased from the American Type Culture Collection (ATCC, Manassas, VA, USA), were cultured in a medium consisting of a MEM/F-12 mixture (HyClone; Cytiva, Marlborough, MA, USA), supplemented with 10% fetal bovine serum (FBS) (HyClone; Cytiva), 1 mM sodium pyruvate (Sigma-Aldrich; Merck KGaA, Darmstadt, Germany), 100 units/mL penicillin (Gibco; Thermo Fisher Scientific, Inc., Waltham, MA, USA), 100 μg/mL streptomycin (Gibco; Thermo Fisher Scientific, Inc.), and 0.1 mM non-essential amino acids (Gibco; Thermo Fisher Scientific, Inc.). These cells were maintained in a humidified incubator at 37 °C with 5% CO_2_ and 95% air, and were harvested for subsequent experiments using a 0.05% trypsin-0.5 mM EDTA solution (Gibco; Thermo Fisher Scientific, Inc.).

### 2.3. H-LIPEF, NAC, and the Phosphoinositide-3-Kinase (PI3K) Inhibitor Treatment

NAC (Sigma-Aldrich; Merck KGaA) was prepared as a 100 mM stock solution in distilled water and stored at −20 °C. SH-SY5Y cells were pretreated with NAC (10 µM) prior to exposure to H-LIPEF for a duration of 4 h. Following this, the cells were challenged with H_2_O_2_ and exposed to NAC and H-LIPEF for an additional 24 h. For the PI3K inhibitor experiment, 10 μM LY294002 was treated 1 h before the NAC and H-LIPEF treatment.

### 2.4. Cell Viability Assay

Following the treatment protocols, the viability of SH-SY5Y cells was assessed using the 3-(4,5-dimethylthiazol-2-yl)-2,5-diphenyltetrazolium bromide (MTT) assay (Sigma-Aldrich; Merck KGaA). Initially, the culture medium was substituted with an MTT solution (0.5 mg/mL in SH-SY5Y culture medium) and incubated at 37 °C for a duration of 4 h. During this incubation, MTT was reduced to formazan crystals by mitochondrial dehydrogenases in living cells, which serve as a marker of mitochondrial activity. Following the MTT reaction, formazan crystals were dissolved using a solution of 10% sodium dodecyl sulfate (SDS) and 0.01 M hydrochloric acid (HCl). The optical density of the resulting solution was then measured at 570 nm, with background correction performed at 690 nm, utilizing an ELISA microplate reader. The percentage of cell viability was subsequently calculated relative to the untreated control, based on the formazan intensity.

### 2.5. Synergy Analysis

The Bliss independence model was used to evaluate interaction between NAC and H-LIPEF on cell viability [42,43]. For calculations, each treatment’s viability was first normalized to the group treated solely with H_2_O_2_. The resulting relative viabilities of the H_2_O_2_ + H-LIPEF, H_2_O_2_ + NAC, and H_2_O_2_ + NAC + H-LIPEF groups are denoted V_H-LIPEF_, V_NAC_, and V_NAC+H-LIPEF_, respectively. If NAC and H-LIPEF act independently, the expected relative viability for the combination is the product V_NAC_ × V_H-LIPEF_. The Bliss synergy score is defined as S_Bliss_ = V_NAC+H-LIPEF_ − (V_H-LIPEF_ × V_NAC_). A positive S_Bliss_ indicates that the observed combination effect exceeds the Bliss expectation, and is interpreted as synergistic protection [43].

### 2.6. Western Blot Analysis

After the treatments, cells were initially washed with ice-cold phosphate-buffered saline (PBS), followed by harvesting and lysis on ice for 1 h in RIPA lysis buffer (EMD Millipore) with a fresh protease and phosphatase inhibitor cocktail (EMD Millipore). The lysates were then subjected to centrifugation at 10,000× *g* at 4 °C for 30 min to separate the cell debris. The supernatants obtained were collected for protein concentration assessment utilizing BSA method. Equal amounts of protein extracts were subsequently loaded into 12% SDS-PAGE wells and transferred onto polyvinylidene fluoride (PVDF) membranes. Following a 1 h blocking period with 5% BSA in TBST washing buffer (composed of 20 mM Tris, 150 mM NaCl, and 0.1% Tween 20), the membranes were incubated overnight at 4 °C with primary antibodies. The specific primary antibodies employed in this study included those targeting p-Akt (Ser473), Nrf2, SOD2, MTH1 (Cell Signaling Technology, Inc., Danvers, MA, USA), poly (ADP-ribose) polymerase (PARP) (cleaved Asp214), and glyceraldehyde 3-phosphate dehydrogenase (GAPDH) (GeneTex, Inc., Irvine, CA, USA) were used. Following three washes with TBST washing buffer, the membranes were incubated with horseradish peroxidase-conjugated secondary antibodies (Jackson ImmunoResearch Laboratories, Inc., West Grove, PA, USA) at room temperature for 1 h. All antibody dilutions were prepared at optimal concentrations according to the manufacturer’s guidelines. Protein bands were visualized using an enhanced chemiluminescence substrate (Advansta, Inc., Menlo Park, CA, USA) and detected with the Amersham Imager 600 imaging system (GE Healthcare Life Sciences, Chicago, IL, USA). The images were analyzed using Image Lab software (version 6.1) (Bio-Rad Laboratories, Inc., Hercules, CA, USA).

### 2.7. Detection of Superoxide Levels

The levels of superoxide in SH-SY5Y cells were assessed using the fluorescent dye dihydroethidium (DHE) (Sigma-Aldrich; Merck KGaA). The cells after treatments were trypsinized, washed with PBS, and subsequently incubated with 5 μM DHE dye for 30 min at 37 °C in the dark. The fluorescence intensity was quantified via flow cytometry in the PE channel. Superoxide levels were expressed as mean fluorescence intensity for comparison.

### 2.8. Intracellular H_2_O_2_ Level Detection

The intracellular levels of H_2_O_2_ in SH-SY5Y cells were measured utilizing the Infra-red Fluorimetric Hydrogen Peroxide Assay Kit (Sigma-Aldrich; Merck KGaA). Following treatment, the cells were subjected to trypsinization, washed with PBS, and incubated with the peroxidase substrate and horseradish peroxidase for 30 min, in accordance with the manufacturer’s instructions. The fluorescence intensity was subsequently quantified using flow cytometry in the APC channel, with H_2_O_2_ levels reported as mean fluorescence intensity for comparative analysis.

### 2.9. Measurement of Mitochondrial Membrane Potential (MMP)

The detection of MMP was performed using flow cytometry with the voltage-dependent, mitochondria-specific lipophilic cationic fluorescent dye 3,3′-dihexyloxacarbocyanine iodide (DiOC_6_(3)) (Enzo Life Sciences, Inc., Plymouth Meeting, PA, USA). Following treatment, SH-SY5Y cells were resuspended in PBS and incubated with 20 nM DiOC_6_(3) for 30 min at 37 °C in the dark. The fluorescence intensity was measured using flow cytometry in the FITC channel to determine the proportion of cells exhibiting low MMP.

### 2.10. Fluorescence Imaging of DNA Oxidative Damage

To detect 8-oxo-2′-deoxyguanosine triphosphate (8-oxo-dG), Alexa Fluor 488-conjugated avidin (Invitrogen; Thermo Fisher Scientific, Inc.) was employed, owing to avidin’s high affinity for 8-oxo-dG. After the treatments, cells were fixed in cold methanol for 20 min at 4 °C, and subsequently subjected to a blocking step using 15% FBS in TBS (20 mM Tris, 150 mM NaCl) containing 0.1% Triton X-100 for 2 h at room temperature. The cells were then incubated with 10 µg/mL of Alexa Fluor 488-conjugated avidin in the blocking solution for 1 h at 37 °C in the dark. Subsequently, the cells were washed three times with TBS containing 0.1% Triton X-100 for a total of 1 h. Finally, the samples were mounted using 4′,6-diamidino-2-phenylindole (DAPI) mounting medium (Abcam, plc., Cambridge, UK) and imaged using a confocal microscope. The fluorescence intensity was quantified using ImageJ software (version 1.54), and the relative intensity of avidin-Alexa 488 was determined by calculating the ratio of the integrated intensity of the green to the blue.

### 2.11. Statistical Analysis

Each data point represents the mean of three independent experiments and is expressed as the mean ± standard deviation. To evaluate statistical significance, one-way analysis of variance (ANOVA) was performed, followed by Tukey’s post hoc test. The statistical analysis was conducted utilizing OriginPro 2015 software (version 9.2) (OriginLab, Northampton, MA, USA). For clarity, only selected pairwise comparisons are annotated on the figures; the full matrices of *p*-values for all pairwise comparisons are provided in Appendix A.

## 3. Results

### 3.1. The Combination of H-LIPEF and NAC Protects SH-SY5Y Cells from H_2_O_2_-Induced Cytotoxicity

Considering the pivotal role of oxidative stress in the progression of NDDs, we exposed SH-SYSY cells to H_2_O_2_ and investigated the neuroprotective effects of H-LIPEF and NAC. As shown in Figure 2A, exposure to H_2_O_2_ for 24 h resulted in a significant reduction in the viability of SH-SY5Y cells in a concentration-dependent manner. Specifically, treatment with 500 µM H_2_O_2_ for 24 h diminished cell viability to 52.5% compared to untreated control cells. Therefore, to induce appropriate oxidative stress in SH-SY5Y cells, 500 µM was used as a standard concentration for subsequent experiments. Next, we evaluated the effects of H-LIPEF and NAC on the viability of SH-SY5Y cells. The parameters for H-LIPEF employed in this study (200 Hz, 10 V/cm, and a pulse duration of 2 ms) were consistent with those utilized in our previous research, which demonstrated optimal protective effects against H_2_O_2_-induced cytotoxicity in SH-SY5Y cells [40]. Regarding NAC treatment, due to its limited bioavailability and uncertain BBB permeability, a relatively low concentration of 10 µM was selected for our experiments, in contrast to the more commonly employed concentrations of 1–10 mM as a ROS scavenger in other in vitro studies [17,44,45,46]. This lower concentration was intended to more accurately reflect the actual levels of NAC present in cerebrospinal fluid in clinical settings [47,48]. For the combination treatment, SH-SY5Y cells were pretreated with NAC and H-LIPEF for 4 h, and then exposed to H_2_O_2_ along with continuous administration of NAC and H-LIPEF for an additional 24 h. As shown in Figure 2B, neither H-LIPEF nor NAC alone, nor their combination, significantly influenced the viability of SH-SY5Y cells compared to the untreated control group. However, when compared to the group treated solely with H_2_O_2_, additional application of H-LIPEF alone and NAC alone increased cell viability by 13.9% and 11.7% (with V_H-LIPEF_ = 1.29 and V_NAC_ = 1.25), respectively. Notably, the NAC and H-LIPEF treatment further enhanced this neuroprotective effect, leading to a 35.5% enhancement in cell viability (with V_NAC+H-LIPEF_ = 1.74). Consequently, the calculated Bliss synergy score (S_Bliss_ = 0.14 > 0) indicates a synergistic protective effect between H-LIPEF and NAC. Additionally, we assessed cell morphology under each experimental condition. As shown in Figure 2C, exposure to H_2_O_2_ caused the SH-SY5Y cells to round and shrink. Interestingly, both NAC and H-LIPEF partially preserved cell morphology, while the combination treatment had the most pronounced effect. Collectively, these findings suggest that H-LIPEF and NAC work synergistically to protect SH-SY5Y cells from H_2_O_2_-induced cytotoxicity.

### 3.2. The Combination of H-LIPEF and NAC Exhibits a Significant Inhibition of H_2_O_2_-Induced Mitochondrial Apoptosis

Mitochondrial dysfunction is a contributing factor to neuronal degeneration [1]. In this study, the MMP of SH-SY5Y cells was assessed using flow cytometry with the DiOC_6_(3) fluorescent dye. As shown in Figure 3A,B, there was a notable increase in the population of cells exhibiting decreased MMP following H_2_O_2_ exposure. Meanwhile, H-LIPEF treatment partially mitigated the loss of MMP induced by H_2_O_2_, and the NAC and H-LIPEF combination further enhanced this protective effect. These findings indicate that NAC and H-LIPEF more effectively protect mitochondria from H_2_O_2_-induced damage. Given that a reduction in MMP is an early indicator of mitochondrial apoptosis, the downstream cleavage of PARP was analyzed via Western blotting. In this study, the ratio of cleaved PARP to full-length PARP serves as a marker for caspase-dependent apoptosis [49]. As shown in Figure 3C, PARP cleavage ratio was significantly increased in H_2_O_2_-treated cells compared to control levels, while H-LIPEF treatment alleviated this effect. Furthermore, the combination of NAC and H-LIPEF further inhibited H_2_O_2_-induced PARP cleavage. Consequently, these results suggest that the combined treatment of NAC and H-LIPEF can significantly prevent mitochondrial apoptosis in SH-SY5Y cells induced by H_2_O_2_.

### 3.3. H-LIPEF and NAC Significantly Attenuate H_2_O_2_-Induced Superoxide Generation

Elevated intracellular ROS levels can lead to mitochondrial dysfunction, ultimately resulting in neuronal apoptosis of neurons in NDDs [50]. Given that superoxide (O_2_^−^) is a primary ROS implicated in NDDs [2], we measured superoxide levels using flow cytometry with DHE fluorescent dye (Figure 4A). Quantification analysis (Figure 4B) revealed that H_2_O_2_ exposure significantly increased superoxide levels, while H-LIPEF treatment partially mitigates this effect. In addition, the combined NAC and H-LIPEF treatment further reduced the superoxide levels to approximately control levels. This suggests that NAC and H-LIPEF can significantly mitigate H_2_O_2_-induced superoxide generation.

### 3.4. The Influence of H-LIPEF and NAC on the p-Akt/Nrf2/SOD2 Signaling Pathway in SH-SY5Y Cells Subjected to H_2_O_2_ Treatment

Next, we investigated the potential signaling pathway that may account for the reduced superoxide levels observed after administering the combined NAC and H-LIPEF treatment. The Akt signaling pathway is recognized for its critical role in neuroprotection. Upon phosphorylation, Akt can activate or stabilize some downstream transcription factors, such as Nrf2, which is responsible for regulating the expression of antioxidant proteins [10]. Therefore, we examined the protein levels of p-Akt using Western blot analysis. As shown in Figure 5A, the level of p-Akt was downregulated upon exposure to H_2_O_2_ compared to untreated control cells; however, this level was significantly restored following H-LIPEF treatment. Furthermore, the combination of NAC and H-LIPEF led to an additional increase in p-Akt levels. Following this, we evaluated the protein levels of the downstream transcription factor Nrf2. As shown in Figure 5B, H_2_O_2_ exposure caused a decrease in Nrf2 levels, while H-LIPEF treatment significantly restored Nrf2 expression, consistent with our previous study [40]. Additionally, NAC and H-LIPEF combination exhibited an even more pronounced effect on enhancing the Nrf2 levels. Moreover, we also assessed the protein levels of the antioxidant enzyme SOD2, which is regulated by Nrf2 and plays a role in catalyzing the conversion of highly reactive superoxide into the more stable H_2_O_2_ [50]. As shown in Figure 5C, exposure to H_2_O_2_ resulted in a slight decrease in SOD2 levels. Compared to the group treated solely with H_2_O_2_, H-LIPEF significantly elevated the expression of SOD2, and the combination of NAC and H-LIPEF further amplified this effect. These findings suggest that the neuroprotective effects of NAC and H-LIPEF may be partially attributed to the activation of the p-Akt/Nrf2/SOD2 signaling pathway, leading to a reduction in the superoxide levels. To further confirm that the neuroprotective effects of NAC and H-LIPEF are mediated through activation of the Akt pathway, we pretreated cells with 10 μM LY294002, an inhibitor of PI3K, the upstream kinase of Akt [51]. As shown in Figure 5D, LY294002 partially abrogated the protective effect of the NAC + H-LIPEF combination on cell viability, supporting the involvement of the Akt pathway in the observed protection.

### 3.5. H-LIPEF and NAC Significantly Attenuate the Increase in the Intracellular H_2_O_2_ Levels

In addition to the conversion of superoxide into H_2_O_2_, we also investigated whether NAC and H-LIPEF could facilitate the further decomposition of H_2_O_2_ into water. In this study, the Infra-red Fluorimetric Hydrogen Peroxide Assay Kit was employed to assess intracellular H_2_O_2_ levels. The results obtained from flow cytometric analysis (Figure 6A) and its quantification (Figure 6B) indicated a significant increase in intracellular H_2_O_2_ levels following exposure to extracellular H_2_O_2_, while treatment with H-LIPEF resulted in a partial reduction in this increase. Notably, the combination of NAC and H-LIPEF led to an even more pronounced decrease in intracellular H_2_O_2_ levels. These findings, in conjunction with previous flow cytometric data on superoxide levels, suggest that the concurrent application of H-LIPEF and NAC effectively facilitates the conversion of superoxide to H_2_O_2_ and promotes the subsequent decomposition of H_2_O_2_ into water, thereby offering substantial protection against oxidative damage in SH-SY5Y cells.

### 3.6. The Combination of H-LIPEF and NAC Demonstrates a Greater Protection of DNA from Oxidative Damage

Oxidative damage to DNA is implicated in neuronal apoptosis and the development of NDDs [52]. One of the predominant forms of oxidative base lesions, 8-oxo-dG, which arises from the oxidation of guanine, serves as a widely recognized biomarker for oxidative DNA damage. This study used avidin-Alexa Fluor 488 and DAPI to visualize 8-oxo-dG and cell nuclei in SH-SY5Y cells, respectively [53]. The representative images obtained are presented in Figure 7A. To quantify the levels of 8-oxo-dG, the relative fluorescence intensity of avidin-Alexa Fluor 488 compared to DAPI was used as an indicator. Figure 7B shows that H_2_O_2_ exposure resulted in an accumulation of 8-oxo-dG, whereas H-LIPEF significantly mitigated this effect. Importantly, the combined treatment of H-LIPEF and NAC exhibited an even more pronounced effect in preventing the accumulation of 8-oxo-dG induced by H_2_O_2_. To investigate the underlying mechanism involved in this effect, the protein levels of MTH1 were assessed, as MTH1 is known to hydrolyze oxidized nucleotides and inhibit their incorporation into DNA [52]. The Western blotting result reveals that MTH1 expression was markedly downregulated following H_2_O_2_ exposure (Figure 7C). In comparison to the group treated solely with H_2_O_2_, H-LIPEF significantly restored MTH1 levels, while the H-LIEPF and NAC combination treatment further enhanced this effect. These results suggest that H-LIPEF and NAC may confer neuroprotective effects by preventing H_2_O_2_-induced oxidative DNA damage through the restoration of MTH1 levels.

## 4. Discussion

The present study aimed to investigate the neuroprotective effect of combining NAC and H-LIPEF in reducing oxidative stress in SH-SY5Y cells. Previous research has established that H-LIPEF can protect SH-SY5Y cells from H_2_O_2_-induced damage [40]. On the other hand, NAC is known for its neuroprotective effects due to its antioxidant properties, as demonstrated in various in vitro experiments [17,44,45,46]. However, the NAC concentrations used in these studies (between 1 and 10 mM) are much higher than those typically found in clinical settings. Specifically, following NAC oral administration, cerebrospinal fluid concentrations of NAC have been reported to peak at approximately 10 µM [47,48]. Therefore, this study chose a low NAC concentration (10 µM) to be physiologically relevant. Our findings, as evidenced by the MTT assay data (Figure 2B), demonstrate that even at this reduced NAC concentration, the combination of NAC and H-LIPEF produces significant neuroprotection against H_2_O_2_-induced cell damage, outperforming the protective effects of either treatment alone. The ability of H-LIPEF to enhance NAC’s protective effects highlights its potential to overcome the issue of NAC’s low bioavailability, representing a significant advancement for clinical applications in NDDs.

To explore the combined neuroprotective effects of NAC and H-LIPEF, this study first evaluated their ability to prevent apoptosis in SH-SY5Y neuronal cells. Neuronal apoptosis can be induced by various stressors, with mitochondrial dysfunction playing a critical role in the onset of apoptosis in NDDs. MMP is an important measure of mitochondrial health, and a decrease in MMP can act as a pivotal switch, triggering the apoptotic cascade [54]. Therefore, maintaining MMP is crucial for preventing apoptosis mediated by mitochondria. Previous research has demonstrated that H-LIPEF can mitigate the loss of MMP in H_2_O_2_-treated SH-SY5Y cells, thereby inhibiting apoptotic signaling pathways [40]. Additionally, many studies indicate that NAC may also improve mitochondrial function [55,56,57,58,59]. Furthermore, this study found that the NAC and H-LIPEF combination treatment had a significantly greater effect in restoring H_2_O_2_-induced MMP depletion than H-LIPEF alone (Figure 3A,B). In addition, activating caspase-dependent pathways is a hallmark of mitochondrial apoptosis. When cells are damaged, mitochondria release cytochrome c from the intermembrane space into the cytosol, which subsequently activates caspases through the formation of the apoptosome complex [60]. This process promotes the activation of executioner caspases that cleave various cellular substrates, including PARP, thereby committing the cell to apoptosis [49]. Our findings indicates that H-LIPEF alone can reduce H_2_O_2_-induced PARP cleavage, while the combination of NAC and H-LIPEF further amplifies this effect (Figure 3C). These results suggest that H-LIPEF and NAC act in combination to inhibit mitochondrial apoptosis induced by H_2_O_2_.

Excessive ROS generation can trigger mitochondrial apoptosis since mitochondria are particularly susceptible to oxidative stress [60]. Among the primary ROS implicated in NDDs are superoxide and H_2_O_2_, with superoxide being the more reactive species and believed to play a crucial role in ROS generation [2,50]. Numerous studies have indicated that H_2_O_2_ treatment can enhance superoxide production [61,62,63,64]. Therefore, targeting superoxide is a vital approach for mitigating cellular damage caused by oxidative H_2_O_2_. Our previous study found that H-LIPEF effectively reduces H_2_O_2_-induced superoxide generation [40]. In this study, we demonstrate that the combination of NAC and H-LIPEF further decreases superoxide levels in H_2_O_2_-treated SH-SY5Y cells, thereby providing protection against oxidative damage (Figure 4). This finding indicates that the anti-apoptotic effects observed with the NAC and H-LIPEF combination may be partially due to their combined inhibitory effect on H_2_O_2_-induced superoxide production.

Regarding the signaling pathways involved, the activation of the Akt pathway appears to be a fundamental mechanism underlying the anti-apoptotic and antioxidative effects in response to oxidative stress. A previous study has shown that NAC pretreatment enhances the expression level of p-Akt, which subsequently protects H9c2 cells from H_2_O_2_-induced apoptosis [17]. On the other hand, our current finding reveals that H-LIPEF alone can upregulate p-Akt protein levels in H_2_O_2_-treated SH-SY5Y cells (Figure 5A). Furthermore, our study indicates that the combination of NAC and H-LIPEF results in a more pronounced increase in p-Akt levels, which may explain the enhanced neuroprotective effects observed with this combined treatment (Figure 5A). Additionally, the activation of the Akt pathway promotes the stabilization of the downstream transcription factor Nrf2, which regulates the expression of various antioxidant proteins and is thus considered a therapeutic target in NDDs [10,11,12,13,14,65]. Our findings indicate that Nrf2 protein levels increase following treatment with NAC and H-LIPEF in H_2_O_2_-treated SH-SY5Y cells, with the combination treatment yielding the most pronounced effect (Figure 5B). One of the key antioxidant proteins regulated by Nrf2 is SOD2, which plays an essential role in mitigating oxidative stress by converting superoxide into the less harmful H_2_O_2_ [50]. Moreover, SOD2 overexpression has been shown to prevent memory deficits in a mouse model of AD [66]. Our study demonstrates that H-LIPEF alone can elevate SOD2 expression in H_2_O_2_-treated SH-SY5Y cells, while the combination of NAC and H-LIEPF exhibits an even greater effect (Figure 5C). These results suggest that the combined treatment of NAC and H-LIPEF may reduce superoxide levels by activating the p-Akt/Nrf2/SOD2 signaling pathway. In addition, the results from the PI3K inhibitor experiment (Figure 5D) further support the involvement of the Akt pathway in the neuroprotective effect provided by the combined treatment of NAC and H-LIPEF, as pretreatment with the inhibitor LY294002 partially attenuated the neuroprotective effect. However, while the Akt pathway’s contribution is significant, our results do not completely exclude the potential involvement of other protective mechanisms. These may include H-LIPEF’s activation of the ERK pathway [40], NAC’s role in increasing intracellular glutathione levels [16], and the MTH1 protein’s action in protecting DNA from oxidative damage (to be discussed later). This is supported by the fact that NAC + H-LIPEF still conferred a minimal restoration of cell viability even in the presence of LY294002.

In addition to the conversion of superoxide to H_2_O_2_, the subsequent decomposition of H_2_O_2_ is critical in preventing Fe^2+^-mediated Fenton reactions that produce highly reactive hydroxyl radicals (•OH), which can cause significant cellular damage [67,68]. Consequently, enhancing the decomposition of H_2_O_2_ has been proposed as a viable therapeutic approach for the treatment of NDDs [69,70]. In this study, we also evaluated intracellular H_2_O_2_ levels in SH-SY5Y cells. The experimental results presented in Figure 6 demonstrate that H-LIPEF treatment partially mitigated the accumulation of intracellular H_2_O_2_, with the combination of H-LIPEF and NAC yielding an even greater effect. This outcome may be attributed to the upregulation of Nrf2 by H-LIPEF and NAC (Figure 5B), as Nrf2 is known to regulate various antioxidant enzymes involved in H_2_O_2_ decomposition [11,12,71,72]. Additionally, NAC has been shown to enhance intracellular glutathione levels, a crucial molecule in the enzymatic reduction of H_2_O_2_ via glutathione peroxidase activity [16]. Collectively, these findings suggest that the combined treatment of NAC and H-LIPEF cooperatively promotes the transformation of superoxide into the less harmful H_2_O_2_ and expedites the decomposition of H_2_O_2_ into water, thereby enhancing the cellular defense against oxidative stress in SH-SY5Y cells.

Maintaining DNA integrity is also vital in NDDs, as oxidative DNA damage can result in cellular dysfunction and apoptosis [52,73]. This investigation assesses the protective effects of NAC and H-LIPEF against H_2_O_2_-induced DNA damage, with a particular emphasis on 8-oxo-dG, a mutagenic lesion that accumulates in the brains of AD patients and contributes to the progression of NDDs [52]. As shown in Figure 7A,B, H-LIPEF alone significantly reduced the levels of 8-oxo-dG in H_2_O_2_-treated SH-SY5Y cells, while the combination of NAC and H-LIPEF demonstrated even greater efficacy, suggesting that this combined treatment could effectively protect DNA from oxidative damage. Furthermore, we evaluated the levels of MTH1 protein, an enzyme that inhibits the incorporation of 8-oxo-dG into DNA. A previous study has reported that MTH1 levels are markedly diminished in the brains of individuals with sporadic AD, which may exacerbate neuroinflammation and increase neuronal death [52]. Additionally, several studies have shown that MTH1-deficient mice are more susceptible to neurodegeneration induced by oxidative stress [74,75,76]. Therefore, restoring MTH1 levels may represent a viable strategy for protecting neurons from oxidative damage and subsequent degeneration. In this study, we found that H-LIPEF alone partially restored MTH1 levels in H_2_O_2_-treated SH-SY5Y cells, while the combination of NAC and H-LIPEF resulted in a more pronounced increase in MTH1 expression (Figure 7C). These results suggest that the combined treatment not only mitigates oxidative DNA damage but also enhances cellular defenses against genotoxic stress by upregulating MTH1, thereby presenting a promising approach for the preservation of neuronal health in NDD patients.

## 5. Conclusions

In conclusion, this study demonstrates that the combined treatment of NAC and H-LIPEF produces a synergistic neuroprotective effect against H_2_O_2_-induced oxidative stress and mitochondrial apoptosis in SH-SY5Y neuronal cells. The underlying mechanisms indicate that this combination effectively mitigates oxidative stress by reducing superoxide levels through the activation of the p-Akt/Nrf2/SOD2 signaling pathway and promoting the conversion of H_2_O_2_ into water, thereby contributing to a robust antioxidative defense mechanism. Furthermore, the findings reveal that the NAC and H-LIPEF combination protects against oxidative DNA damage by decreasing the levels of 8-oxo-dG and enhancing the expression of MTH1. Notably, the capacity of H-LIPEF to amplify the effects of low NAC concentrations underscores its potential to address the limitations associated with NAC’s low bioavailability, presenting a promising therapeutic strategy for the treatment of NDDs. Further research is necessary to investigate the actual electric field strength within biological tissues, as well as factors such as dielectric shielding and field distribution, to better elucidate the in vivo feasibility and safety of H-LIPEF. Moreover, future studies should focus on optimizing the parameters of H-LIPEF and exploring its application in combination with other neuroprotective agents, with the ultimate goal of translating these findings into clinical applications for the management of NDDs.

## Figures and Tables

**Figure 1 antioxidants-14-01267-f001:**
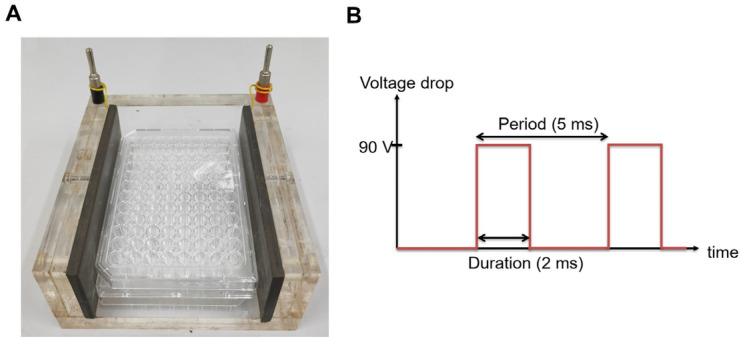
The experimental setup of high-frequency, low-intensity pulsed electric field (H-LIPEF). (**A**) A photograph of the H-LIPEF stimulation apparatus. (**B**) The waveform of the voltage drop applied across the two electrodes.

**Figure 2 antioxidants-14-01267-f002:**
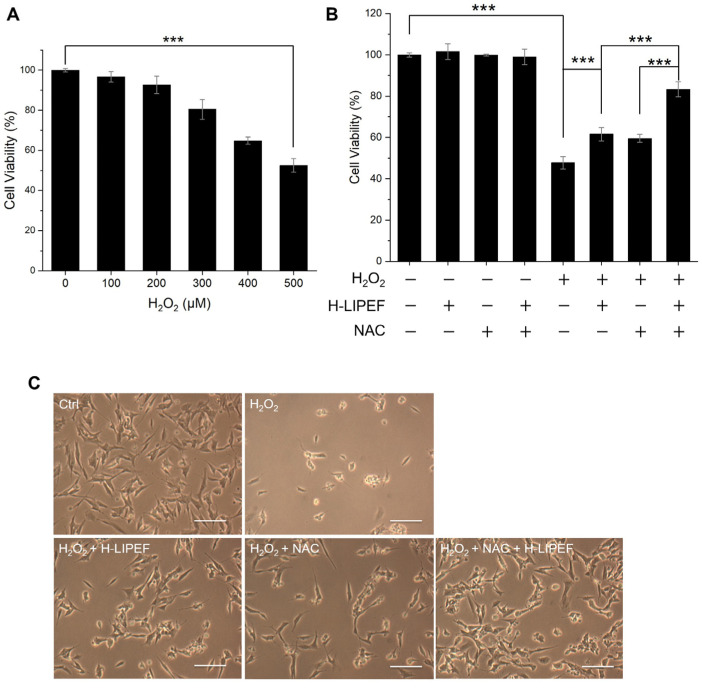
Effect of H-LIPEF and N-acetylcysteine (NAC) alone or their combinational treatment on H_2_O_2_-induced cytotoxicity in SH-SY5Y cells. (**A**) SH-SY5Y cells were challenged with different concentrations of H_2_O_2_. The cell viability was measured by 3-(4,5-dimethylthiazol-2-yl)-2,5-diphenyltetrazolium bromide (MTT) assay 24 h after the initiation of H_2_O_2_ exposure. (**B**) The viability of SH-SY5Y cells after H-LIPEF, NAC (10 μM), H_2_O_2_ (500 μM) either alone, in combination with each other, or in triple combination. The cell viability was measured by MTT assay 24 h after the initiation of H_2_O_2_ exposure. (**C**) Representative light microscopy images of SH-SY5Y cells after H_2_O_2_, H_2_O_2_ + H-LIPEF, H_2_O_2_ + NAC, and H_2_O_2_ + NAC + H-LIPEF treatments. The images were taken 24 h after the initiation of H_2_O_2_ exposure. The exposure to H_2_O_2_ resulted in significant cellular damage, whereas the combination treatment of H-LIPEF and NAC demonstrated the most pronounced protective effect, preserving cell morphology. Scale bar = 50 μm. *** *p* < 0.001, comparison between indicated groups.

**Figure 3 antioxidants-14-01267-f003:**
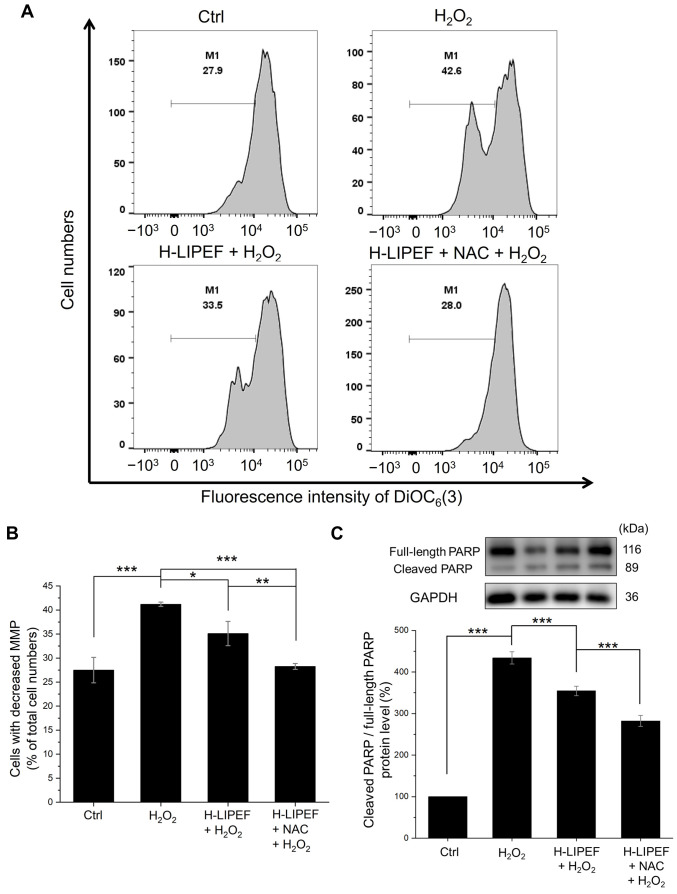
Effects of H-LIPEF alone or in combination with NAC on H_2_O_2_-induced mitochondrial apoptosis in SH-SY5Y cells. (**A**) Flow cytometric analysis of mitochondrial membrane potential (MMP) using 3,3′-dihexyloxacarbocyanine iodide (DiOC_6_(3)) staining. (**B**) Quantification of the percentage of cells exhibiting decreased MMP (M1 regions in (**A**)) following treatments with H_2_O_2_, H-LIPEF + H_2_O_2_, and H-LIPEF + NAC + H_2_O_2_. (**C**) Representative images of Western blotting and quantification of the ratio between cleaved poly (ADP-ribose) polymerase (PARP) and full-length PARP protein levels in SH-SY5Y cells after treatments with H_2_O_2_, H-LIPEF + H_2_O_2_, and H-LIPEF + NAC + H_2_O_2_. Each relative expression level was compared to the control. Statistical significance is indicated as * *p* < 0.05, ** *p* < 0.01, and *** *p* < 0.001, in comparisons between the indicated groups.

**Figure 4 antioxidants-14-01267-f004:**
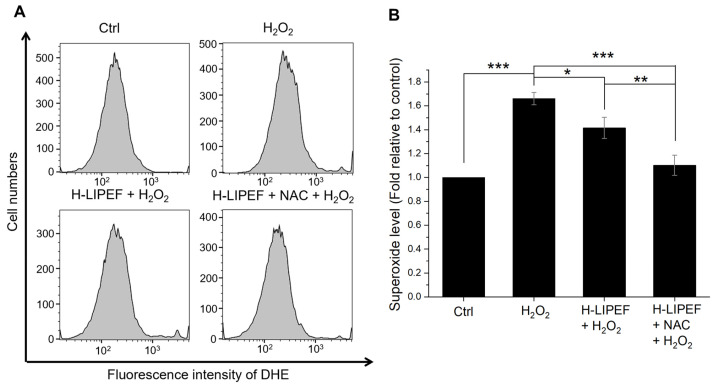
Effects of H-LIPEF alone or in combination with NAC on the production of superoxide induced by H_2_O_2_ in SH-SY5Y cells. (**A**) Flow cytometric analysis of superoxide levels using dihydroethidium (DHE) staining. (**B**) Quantification of the superoxide levels following treatments with H_2_O_2_, H-LIPEF + H_2_O_2_, and H-LIPEF + NAC + H_2_O_2_ in comparison to the control group. Statistical significance is indicated with * *p* < 0.05, ** *p* < 0.01, and *** *p* < 0.001 for comparisons between the indicated groups.

**Figure 5 antioxidants-14-01267-f005:**
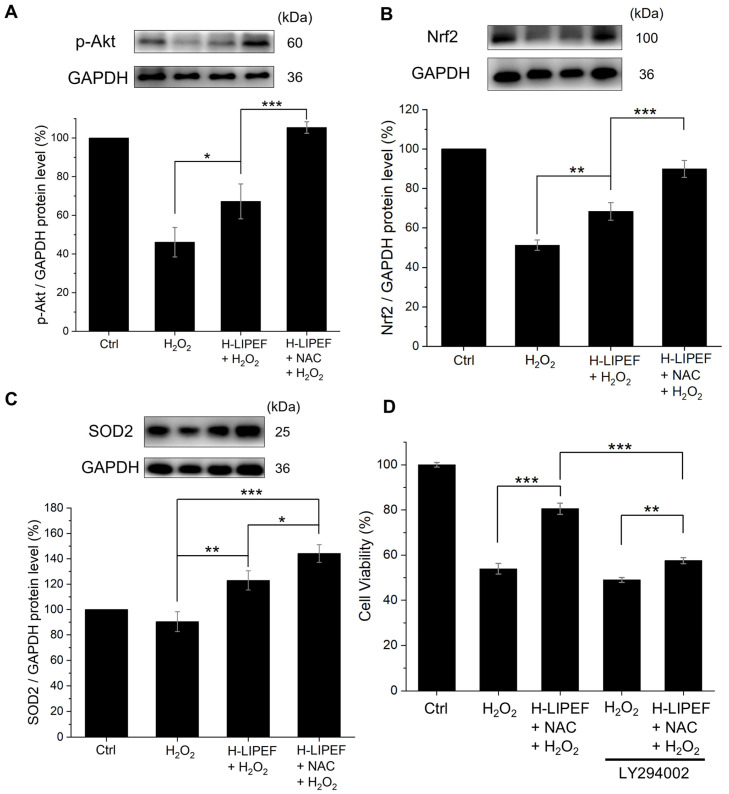
Effects of H-LIPEF alone or in combination with NAC on the phosphorylated Akt (p-Akt)/nuclear factor erythroid 2-related factor 2 (Nrf2)/superoxide dismutase type 2 (SOD2) signaling pathway in SH-SY5Y cells. Panels (**A**), (**B**), and (**C**) present representative Western blot images and quantification of p-Akt, Nrf2, and SOD2 protein levels, respectively, following treatments with H_2_O_2_, H-LIPEF + H_2_O_2_, and H-LIPEF + NAC + H_2_O_2_. Panel (**D**) shows the cell viability assessed by MTT assay, demonstrating that H-LIPEF + NAC increases cell viability under H_2_O_2_ stress. This neuroprotective effect was partially abrogated by pretreatment with the phosphoinositide-3-kinase (PI3K) inhibitor, LY294002. Statistical significance is denoted with * *p* < 0.05, ** *p* < 0.01, and *** *p* < 0.001 for comparisons between the indicated groups.

**Figure 6 antioxidants-14-01267-f006:**
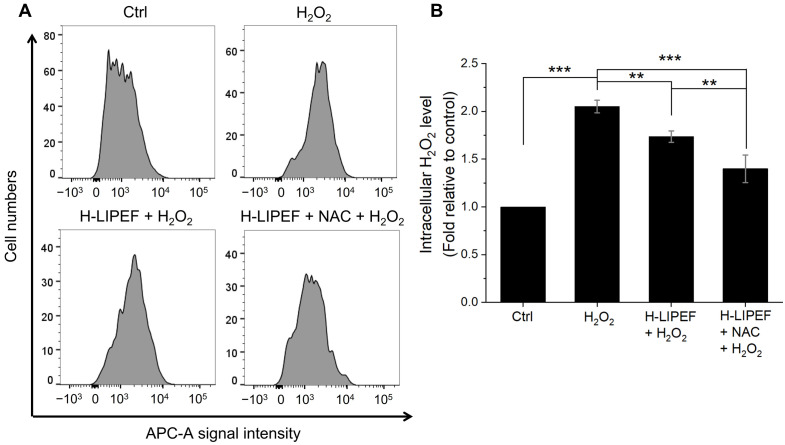
Effects of H-LIPEF alone or in combination with NAC on intracellular H_2_O_2_ levels in SH-SY5Y cells. (**A**) Flow cytometric analysis of intracellular H_2_O_2_ level using Infra-red Fluorimetric Hydrogen Peroxide Assay Kit. (**B**) Quantification of the intracellular H_2_O_2_ levels after treatments with H_2_O_2_, H-LIPEF + H_2_O_2_, and H-LIPEF + NAC + H_2_O_2_ in comparison to the control group. Statistical significance is indicated as ** *p* < 0.01 and *** *p* < 0.001, in comparisons between the indicated groups.

**Figure 7 antioxidants-14-01267-f007:**
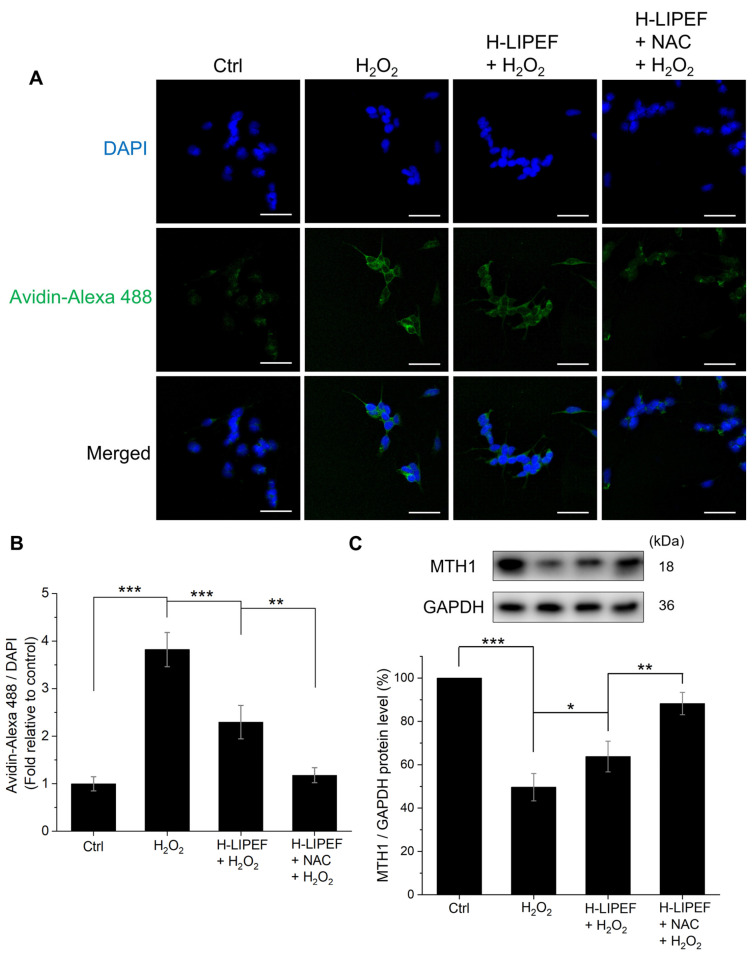
Effects of H-LIPEF alone or in combination with NAC on DNA protection in SH-SY5Y cells. (**A**) Representative confocal images of SH-SY5Y cells stained with avidin-Alexa Fluor 488 conjugate and 4′,6-diamidino-2-phenylindole (DAPI). Scale bar = 50 μm. (**B**) The accumulation of 8-oxo-2′-deoxyguanosine triphosphate (8-oxo-dG) was quantified by calculating the ratio of fluorescence intensity of avidin-Alexa Fluor 488 (green) to DAPI (blue). (**C**) Representative images of Western blotting and the quantification of MutT homolog 1 (MTH1) protein levels following treatment with H_2_O_2_, H-LIPEF + H_2_O_2_, and H-LIPEF + NAC + H_2_O_2_ in comparison to the control group. Statistical significance is indicated as * *p* < 0.05, ** *p* < 0.01, and *** *p* < 0.001 for comparisons between the indicated groups.

## Data Availability

The data presented in this study are available on request from the corresponding author.

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
