# Peer review of "High-Frequency, Low-Intensity Pulsed Electric Field and N-Acetylcysteine Synergistically Protect SH-SY5Y Cells Against Hydrogen Peroxide-Induced Cell Damage In Vitro"

_antioxidants, 2025, doi:10.3390/antiox14101267_

Round 1
Reviewer 1 Report
Introduction : While the Introduction provides a comprehensive background on oxidative stress and the therapeutic potential of NAC in neurodegenerative diseases, the rationale for focusing on the p-Akt/Nrf2/SOD2 signaling pathway is not sufficiently explained. The manuscript changed suddenly from discussing ROS and NAC to mentioning Nrf2/SOD2, with only a brief connection to Akt through prior LIPEF studies. For clarity and coherence, it would be beneficial to introduce this pathway explicitly. Establishing this mechanistic context at the outset would create a stronger logical foundation for the study’s hypothesis and subsequent experimental focus.
NAC Concentration: The rationale for using 10 μM NAC is commendable, as it reflects cerebrospinal fluid concentrations observed in clinical studies. However, no direct comparison with commonly used higher in vitro concentrations (1–10 mM) is provided. Without such comparisons, the interpretation of synergistic effects remains limited.
Safety of H-LIPEF: While the manuscript states that the applied electrical parameters are within safety standards, the discussion does not adequately address potential safety and penetration issues in vivo. This aspect requires further elaboration to strengthen the translational perspective.
Mechanistic Insights(Discussion) : The mechanistic focus of the study remains somewhat ambiguous. While the authors report activation of the p-Akt/Nrf2/SOD2 pathway in parallel with the observed protective effects, it is not clear whether this pathway is the primary target or merely associated with the treatment response. Establishing a more rational causal relationship—such as by using pathway inhibitors, knockdown approaches, or rescue experiments—would greatly strengthen the mechanistic interpretation. At minimum, this limitation should be clearly acknowledged in the Discussion.
Word : The repeated use of the word “synergistic” in the Abstract and Discussion might be overstated, as some of the effects appear additive rather than strictly synergistic. A more cautious wording is recommended.
Abbreviation : Abbreviations (e.g., NAC, H-LIPEF) are properly defined initially but inconsistently reintroduced in later sections, which may cause confusion for readers.
Statistical analysis: In Figures 2B and subsequent experiments, the Hâ‚‚Oâ‚‚-treated group appears to serve as the reference control against which other treatments are compared. In this context, Dunnett’s test may be more appropriate than Tukey’s test, which assumes pairwise comparisons among all groups. If Tukey was used, the authors should explain why only selected pairwise comparisons are presented and ensure that the statistical approach is fully consistent with the way the results are displayed.
Author Response
Dear Reviewer 1,
We are grateful for your insightful review of our manuscript. Your comments have provided us with significant guidance for improving the quality and clarity of our research. In accordance with your suggestions, we have meticulously revised the manuscript to improve its scientific rigor, clarity, and translational significance.
The following is a point-by-point response to your comments.
Comment 1:
Introduction : While the Introduction provides a comprehensive background on oxidative stress and the therapeutic potential of NAC in neurodegenerative diseases, the rationale for focusing on the p-Akt/Nrf2/SOD2 signaling pathway is not sufficiently explained. The manuscript changed suddenly from discussing ROS and NAC to mentioning Nrf2/SOD2, with only a brief connection to Akt through prior LIPEF studies. For clarity and coherence, it would be beneficial to introduce this pathway explicitly. Establishing this mechanistic context at the outset would create a stronger logical foundation for the study’s hypothesis and subsequent experimental focus.
Response 1:
We appreciate this constructive suggestion. We have revised the Introduction to explicitly explain the p-Akt/Nrf2 signaling pathway and its role in antioxidant defense in the second paragraph. Additionally, we highlighted reports in the subsequent paragraph indicating that NAC can activate Akt signaling, with relevant references cited, thereby providing a clearer mechanistic rationale for our study.
The revisions appear on page 2, lines 56-63 and lines 67-68.
Comment 2:
NAC Concentration: The rationale for using 10 μM NAC is commendable, as it reflects cerebrospinal fluid concentrations observed in clinical studies. However, no direct comparison with commonly used higher in vitro concentrations (1–10 mM) is provided. Without such comparisons, the interpretation of synergistic effects remains limited.
Response 2:
We thank the reviewer for this insightful comment. Indeed, we know that concentration–response gradients are typically required for most synergy evaluation methods. However, in our experimental system, this approach is not readily applicable because the physical stimulation by H-LIPEF does not have a well-defined “dosage” comparable to chemical concentrations. Moreover, our primary aim was to examine whether a physiologically relevant NAC concentration (10 μM) could exert a measurable neuroprotective effect when combined with H-LIPEF.
To evaluate the interaction between NAC and H-LIPEF, we have applied the Bliss independence model using cell viability data [1, 2]. This quantitative approach enables us to determine whether NAC and H-LIPEF exhibits a synergistic protection effect, without the need for a concentration–response series of NAC. The synergy analysis using the Bliss independence model has been incorporated into the revised manuscript, appearing on page 5, lines 171-179 and page 7, lines 269-273.
Comment 3:
Safety of H-LIPEF: While the manuscript states that the applied electrical parameters are within safety standards, the discussion does not adequately address potential safety and penetration issues in vivo. This aspect requires further elaboration to strengthen the translational perspective.
Response 3:
We thank the reviewer for this important point. We have noted that the non-contact delivery of H-LIPEF avoids adverse sensations such as itching, tingling, or skin lesions, and the exposure parameters used in this study fall within currently accepted international safety guidelines. Importantly, we observed that H-LIPEF alone produced no detectable cytotoxicity in SH-SY5Y cells under the applied exposure conditions, which further supports the view that H-LIPEF is unlikely to cause direct harm to human cells.
However, we acknowledge that several critical issues must still be addressed to translate our in vitro findings to in vivo applications. Accordingly, we have expanded the Discussion to more fully address these issues, including the need to determine actual electric-field strength within biological tissues, the effects of dielectric shielding and field distribution.
These additions are included in the final paragraph of the Discussion (pages 17-18, lines 555-557).
Comment 4:
Mechanistic Insights(Discussion) : The mechanistic focus of the study remains somewhat ambiguous. While the authors report activation of the p-Akt/Nrf2/SOD2 pathway in parallel with the observed protective effects, it is not clear whether this pathway is the primary target or merely associated with the treatment response. Establishing a more rational causal relationship—such as by using pathway inhibitors, knockdown approaches, or rescue experiments—would greatly strengthen the mechanistic interpretation. At minimum, this limitation should be clearly acknowledged in the Discussion.
Response 4:
We appreciate the reviewer’s insightful comment. We fully agree that establishing a more definitive mechanistic relationship is crucial for the interpretation of our findings. To address this critical point, we have performed additional experiments using the PI3K inhibitor LY294002, which is known to block the phosphorylation of Akt [3]. The results (now included as Figure 5D in the revised manuscript) clearly show that pre-treatment with the LY294002 partially attenuates the protective effects of NAC + H-LIPEF. This finding supports that the activation of the PI3K/Akt pathway is a critical, causal event (rather than merely an association) in mediating the observed protective effects. We have incorporated the description, results, and interpretation of this experiment into the revised manuscript. This content appears on page 4, lines 155-157; page 11, lines 360-365; page 12, lines 371-374; and page 16, lines 496-505.
Comment 5:
Word : The repeated use of the word “synergistic” in the Abstract and Discussion might be overstated, as some of the effects appear additive rather than strictly synergistic. A more cautious wording is recommended.
Response 5:
We thank the reviewer for this important comment. We agree that the term “synergistic” may overstate some of the results. We also acknowledge that, aside from cell viability, rigorous synergy analyses were not performed for the other experimental results. Accordingly, we have replaced some occurrences of “synergistic” in the manuscript with more cautious terms appropriate to each context.
These wording changes have been incorporated into the revised manuscript, as indicated on page 7, lines 279 and 295; page 10, lines 321 and 329; page 13, line 376; page 15, line 406; page 16, lines 462 and 473-474; page 17, line 519.
Comment 6:
Abbreviation : Abbreviations (e.g., NAC, H-LIPEF) are properly defined initially but inconsistently reintroduced in later sections, which may cause confusion for readers.
Response 6:
We thank the reviewer for pointing out this potential inconsistency. Our handling of abbreviations follows the specific guidelines provided in the "Instructions for Authors" for the journal Antioxidants, which states: "Acronyms/Abbreviations/Initialisms should be defined the first time they appear in each of three sections: the abstract; the main text; and the first figure or table. When defined for the first time, the acronym/abbreviation/initialism should be added in parentheses after the written-out form." In accordance with this rule, we have carefully re-examined the entire manuscript and confirmed that all abbreviations are correctly defined upon their first use in each of these required sections. These revisions can be seen throughout the revised manuscript.
Comment 7:
Statistical analysis: In Figures 2B and subsequent experiments, the Hâ‚‚Oâ‚‚-treated group appears to serve as the reference control against which other treatments are compared. In this context, Dunnett’s test may be more appropriate than Tukey’s test, which assumes pairwise comparisons among all groups. If Tukey was used, the authors should explain why only selected pairwise comparisons are presented and ensure that the statistical approach is fully consistent with the way the results are displayed.
Response 7:
Thank you for the helpful comment regarding the choice of multiple-comparison test. We agree that Dunnett’s test is appropriate when the sole objective is to compare all treatment groups to a single control (e.g., Hâ‚‚Oâ‚‚). However, in this study we needed to perform direct pairwise comparisons not only versus the Hâ‚‚Oâ‚‚ damage control, but also between combination and single-treatment groups (for example, NAC+Hâ‚‚Oâ‚‚ vs NAC+H-LIPEF+Hâ‚‚Oâ‚‚, and/or H-LIPEF+Hâ‚‚Oâ‚‚ vs NAC+H-LIPEF+Hâ‚‚Oâ‚‚) to determine whether the combined treatment differs significantly from each constituent treatment, therefore we have used Tukey’s test. The reason that only selected pairwise comparisons are annotated in the figures is for visual clarity (this explanation is provided on page 6, lines 241-243 of the manuscript) and to emphasize that the combination of NAC and H-LIPEF exhibits a greater protective effect than either treatment alone. To improve transparency and to address the reviewer’s concern, we have provided Supplementary Table S1 containing the full matrix of adjusted p-values for all pairwise comparisons obtained from Tukey’s test.
References:
- Pearson, R.A.; Wicha, S.G.; Okour, M. Drug combination modeling: methods and applications in drug development. Clin. Pharmacol. 2023, 63, 151–165.
- Kang, Y.; Tierney, M.; Ong, E.; Zhang, L.; Piermarocchi, C.; Sacco, A.; Paternostro, G. Combinations of kinase inhibitors protecting myoblasts against hypoxia. PLoS ONE 2015, 10, e0126718.
- Moreira, N.C.D.S.; Piassi, L.O.; Lima, J.E.B.F.; Passos, G.A.; Sakamoto-Hojo, E.T. PTEN inhibition induces neuronal differentiation and neuritogenesis in SH-SY5Y cells via AKT signaling pathway. Alzheimers Dis. 2025, 106, 1436–1451.
Finally, we would like to thank you again for your valuable comments. We believe that these revisions have significantly enhanced the manuscript and improved its overall quality.

Reviewer 2 Report
The article “High-frequency, low-intensity pulsed electric field and N-acetylcysteine synergistically protect SH-SY5Y cells against hydrogen peroxide-induced cell damage in vitro” investigated the synergistic neuroprotective effects of combining N-acetylcysteine (NAC) with non-contact high-frequency low-intensity pulsed field (H-LIPEF) stimulation on neuronal cells. The combined treatment significantly improved cell viability, preserved cell morphology, reduced oxidative stress, activated the p-Akt/Nrf2/SOD signaling pathway, and protected against DNA damage in SH-SY5Y human neuronal cells.
This is a very interesting topic, especially since neurodegenerative diseases often cause a gradual decline in cognitive or motor functions. These results suggest that combined treatment with NAC and H-LIPEF enhances cellular defenses against genotoxic stress and increases the neuroprotective efficacy of low-dose NAC. Therefore, the findings have important potential for improving therapeutic strategies for neurodegenerative diseases.
Adequate methodologies were used.
The results are presented in seven clear and well-organized figures.
The literature should be updated, as only one-third of the references are from the last five years.
Author Response
Dear Reviewer 2,
We are very pleased that the reviewer found our study on the synergistic neuroprotective effects of NAC and H-LIPEF to be highly interesting and of significant potential. We appreciate the positive comments regarding our methodology, figures, and the potential clinical significance for improving therapeutic strategies in neurodegenerative diseases.
The following is a response to your detailed comment.
Comment:
The literature should be updated, as only one-third of the references are from the last five years.
Response:
We thank the reviewer for this comment. We acknowledge the comment regarding the need to update our literature. To address this, we have conducted a thorough search and have added new and highly relevant references published within the last five years (now cited as references 3, 5, 11-15, 18, 21, 24, 42, 51, 59 in the revised manuscript). These additions ensure our discussion is supported by the most current scientific evidence, thereby strengthening the manuscript.
Thank you once again for the time and effort spent in reviewing our manuscript.

Round 2
Reviewer 1 Report
-
The authors’ revisions and explanations are satisfactory. I believe this version addresses the previous concerns, and I wish them the best with the manuscript.